# Glutaredoxin Interacts with GR and AhpC to Enhance Low-Temperature Tolerance of Antarctic Psychrophile *Psychrobacter* sp. ANT206

**DOI:** 10.3390/ijms23031313

**Published:** 2022-01-24

**Authors:** Yatong Wang, Quanfu Wang, Yanhua Hou, Jianan Liu

**Affiliations:** 1School of Environment, Harbin Institute of Technology, Harbin 150090, China; wangyatong199311@163.com; 2School of Marine Science and Technology, Harbin Institute of Technology, Weihai 264209, China; liujianan0623@163.com

**Keywords:** antarctic, glutaredoxin, deletion mutant, oxidative stress, yeast two-hybrid, BiFC

## Abstract

Glutaredoxin (Grx) is an important oxidoreductase to maintain the redox homoeostasis of cells. In our previous study, cold-adapted Grx from *Psychrobacter* sp. ANT206 (PsGrx) has been characterized. Here, we constructed an in-frame deletion mutant of *psgrx* (Δ*psgrx*). Mutant Δ*psgrx* was more sensitive to low temperature, demonstrating that *psgrx* was conducive to the growth of ANT206. Mutant Δ*psgrx* also had more malondialdehyde (MDA) and protein carbonylation content, suggesting that PsGrx could play a part in the regulation of tolerance against low temperature. A yeast two-hybrid system was adopted to screen interacting proteins of 26 components. Furthermore, two target proteins, glutathione reductase (GR) and alkyl hydroperoxide reductase subunit C (AhpC), were regulated by PsGrx under low temperature, and the interactions were confirmed via bimolecular fluorescence complementation (BiFC) and co-immunoprecipitation (Co-IP). Moreover, PsGrx could enhance GR activity. *trxR* expression in Δ*psgrx*, Δ*ahpc,* and ANT206 were illustrated 3.7, 2.4, and 10-fold more than mutant Δ*psgrx* Δ*ahpc*, indicating that PsGrx might increase the expression of *trxR* by interacting with AhpC. In conclusion, PsGrx may participate in glutathione metabolism and ROS-scavenging by regulating GR and AhpC to protect the growth of ANT206. These findings preliminarily suggest the role of PsGrx in the regulation of oxidative stress, which could improve the low-temperature tolerance of ANT206.

## 1. Introduction

Glutaredoxin (Grx), which is widely distributed in the cells of bacteria, plants, and mammals, is a general glutathione–disulfide reductase of importance in redox regulation. Grxs can be broadly separated into two highly abundant major subfamilies, which are termed class I and II Grxs. Those in class I present the oxidoreductase activity, which control a variety of protein thiol redox homeostasis; these typically include dithiol enzymes with two active-site cysteine residues [1]. A class II Grxs play a role in regulating iron (Fe) metabolism as well as the maturation of the iron–sulfur protein [2,3]. Reversible redox modification of proteins is considered to be an important regulatory mechanism in organisms. Many signal molecules and transcription factors function through changes in the redox state of proteins [4]. Adverse stress often triggers the production of reactive oxygen species (ROS) in an organism, which changes the redox state in the cell. As an important thiol disulfide bond oxidoreductase in the cell, Grxs play a significant part in the regulation of the intracellular redox balance and the process of resisting oxidative stress damage, which has become a hot scientific topic [5]. In recent years, Grxs from plants have emerged as key regulators during stress. The transcript levels of rice Grx20 significantly respond to salt treatment [6]. Brassinosteroid-mediated apoplastic H_2_O_2_-glutaredoxin cascade regulates antioxidant capacity in response to chilling in tomato [7]. In addition, the overexpression of an *Arabidopsis* monothiol glutaredoxin AtGRXS17 improves response to heat stress in chrysanthemum [8]. Iron–sulfur GrxS17 activates holdase activity and protects plants from heat stress [9]. Furthermore, Grxs from other organisms also possess ability to resistance stress. Grx from *Synechocystis* sp. PCC 6803 may provide protection to *E. coli* cells against oxidative stress [10]. Grx from yeast *Saccharomyces cerevisiae* is required for protection against ROS [11]. Grx in yeast plays distinct roles during normal cellular growth and in response to stress conditions [12], and the disruption of Grx causes oxidative damage and growth defects of *Candida albicans* [13]. Moreover, human Grx has the potential to regulate iron homeostasis via delivery of a cluster to iron regulatory proteins [14]. However, the regulatory mechanism of bacterial-derived Grxs under oxidative stress is still unclear.

Antarctica is considered the driest, windiest, and coldest place on Earth; it is completely isolated, geographically and thermally, from the other continents [15]. Biological systems in Antarctica are unavoidably affected by changes in ambient temperature, which interfere with the state of metabolism and cellular signal processing [16]. In fact, the drastic changes in temperature in Antarctic habitats cause the production of hydroxyl free radicals and ROS, which have a negative impact on the survival of microbial cells, such as damaging proteins, lipids, and DNA, resulting in cell death [17,18]. Antioxidant enzymes start to work to remove ROS, and they are gradually being tapped. Glutathione S-transferase from Antarctic bacteria shows its protective effects against oxidative stresses [19]. Superoxide dismutase and ascorbate peroxidase from Antarctic microorganisms have also been identified [20]. The psychrophile *Psychrobacter*, a typical strain in Antarctica, contains antioxidant enzymes such as nitroreductase and glutathione reductase [21,22]. Recently, Grx from Antarctic *Psychrobacter* sp. ANT206 (PsGrx) was reported [23]. The optimal temperature for PsGrx activity is 25 °C, and the enzyme retains almost 40% residual activity at even 0 °C, demonstrating that it is a cold-adapted enzyme. Importantly, PsGrx protects supercoiled DNA from oxidation-induced damage under low temperature. The objective of the present study was to further elucidate the functions and interactions of PsGrx from *Psychrobacter* sp. ANT206 in low temperature. We constructed an in-frame deletion mutant strain of *psgrx*, *ahpc,* and double deletion mutant, yeast two-hybrid (Y2H) and bimolecular fluorescence complementation (BiFC) were applied to test and verify the target proteins of PsGrx. The results suggest a function of Grx in microbial cold adaptation from the perspective of renewal.

## 2. Results and Discussion

### 2.1. Construction and Analysis of the Deletion Mutant Δpsgrx

According to the schematic diagram of the in-frame gene deletion mutant strain construction process of *psgrx* in ANT206 (Figure 1), the Δ*psgrx* gene sequence (data not shown), and the results shown in Figure 2A, a segment of 90 bp was deleted in *psgrx* ORF, and an in-frame deletion mutation gene with a length of 174 bp was identified. Subsequently, the mutant Δ*psgrx* was successfully constructed via homologous recombination and screened by 10% sucrose sensitivity and kanamycin resistance. The mutant Δ*psgrx* was capable of inheriting more than 30 generations (Figure 2B), indicating that it possessed good hereditary and stable kanamycin resistance.

### 2.2. PsGrx Positively Regulates the Response to Low Temperature

This study further examined the differences in the growth values of ANT206 and the mutant Δ*psgrx* under low temperature (Figure 3A). The cell density values of wild-type ANT206 and mutant Δ*psgrx* were similar under normal culture conditions (15 °C), indicating that *psgrx* was not essential to strain ANT206 survival. Compared to ANT206, the mutant Δ*psgrx* exhibited a slower growth rate under low temperature, suggesting that it was more sensitive to temperature than the wild type. The maximum specific growth rates (μ_max_) and generation time (GT) of mutant Δ*psgrx* compared with ANT206 for each condition was illustrated in Figure 3B,C. It can be seen that μ_max_ and GT were affected by temperature; these two strains showed a similar growth rate at 15 °C, and the value of μ_max_ was higher in ANT206 than mutant Δ*psgrx* at low temperatures, indicating that *psgrx* deletion impaired low-temperature growth. Furthermore, a previous study had used the value of μ_max_ to evaluate the growth of mutant strains and analyzed the effect of deleted genes of *Saccharomyces cerevisiae* under low temperature [24]. Meanwhile, the μ_max_ of the *Saccharomyces strains* grown at 15 °C was also analyzed [25]. The above findings indicated that *psgrx* was conducive to the growth of ANT206 under low temperature.

MDA is an end product of oxygen free radicals reacting with unsaturated fatty acids in the cell membrane, and it is commonly employed as a marker of lipid peroxidation [26]. Its content can reflect the severity of cell damage; as the cells are damaged by ROS, the content of MDA increased [27]. Similarly, protein carbonylation content is regarded as a biomarker of oxidative stress [28]. Low-temperature treatment increased the MDA content both in ANT206 and the mutant Δ*psgrx* (Figure 3D), which is the phenomenon reported for wheat leaves [29]. Furthermore, the content of MDA and protein carbonylation content were higher in the deletion mutant Δ*psgrx* (Figure 3E), indicating that the deletion of *psgrx* could cause an imbalance of oxidative metabolism in the bacteria, thus weakening the tolerance to low temperature. Taken together, these results indicated that *psgrx* was considered to be involved in low-temperature regulation and acted as a positive regulator of low-temperature tolerance to ANT206.

### 2.3. Screening for Proteins That Interact with PsGrx

In order to understand the mechanism of how PsGrx regulated the low-temperature stress response in ANT206, the relationship between the *psgrx* and the target proteins was studied. The yeast two-hybrid system is an effective assay for studying protein–protein interactions [30,31]. Here, it was used to find proteins that potentially bind to PsGrx. Colony counts of 1440, 242, and 22 were successfully transformed and grown on plates with dilutions of 1:10, 1:100, and 1:1000, respectively. The transformation efficiency was calculated to be 3.23 × 10^4^/ug. The average insertion length of the library was about 1.2 kbp (Appendix A), the positive clone rate exceeded 95%, and the library capacity was 1.15 × 10^7^ CFU. 

Autoactivation was tested before screening with *psgrx* as bait. As shown in Figure 4, strains grew on SD-TL-deficient plates. Six colonies of each group were randomly selected and transferred to SD-TLHA+X-α-Gal-defective plates. The pGADT7 + pGBKT7-PsGrx and negative control did not grow, while the positive control grew. Taken together, these results confirmed that pGADT7 and pGBKT7-PsGrx did not activate the reporter genes (HIS3, ADE2 and MEL1) autonomously when expressed in Y2H Gold yeast cells, suggesting that the plasmids were suitable for using in Y2H screening.

Thirty blue colonies were obtained through Y2H screening (Figure 5). The complete sequences of 26 colonies were obtained. The putative targets, including stress response, translation, Calvin cycle, sulfur metabolism, nitrogen metabolism, protein secretion, RNA metabolism, and protein synthesis, to name a few, are listed in Table 1. Glutathione reductase (GR), a member of the Grx system, plays roles in oxidative stress [32], as does glutathione peroxidase. Alkyl hydroperoxide reductase (AhpC) scavenges a variety of peroxides, ROS, and nitrogen and sulfur species [33]. Proteins relate to the Calvin cycle and associated reactions are also targeted by Grx and Trx in plants [34,35]. The sulfur metabolism-related protein methionine synthase and cysteine synthase were also screened. Among these potential interacting proteins, there were several new targets of Grx. GspI is involved in protein secretion across the membrane in Gram-negative bacteria [36]. Furthermore, the DNA translocase FtsK helps coordinate cell division with DNA unlinking and segregation, which influences cell cycle regulation. RNase E, which functions in the degradation of mRNA, is a member of the RNase E/G family protein. In *Escherichia coli*, it is involved in the dominant pathways of mRNA transcript decay and RNA metabolism [37,38]. Protein synthesis involves the AarF/Abc1/UbiB kinase family proteins. Above all, the interactions between these proteins and PsGrx suggest that PsGrx might play a vital role in the regulation of biological processes. All proteins identified will not be described here, but emphasis will be made about stress-related reactions and new targets. Next, stress-related targets (GR, glutathione peroxidase and AhpC) and new targets (GspI, FtsK and RNase E) were selected for further identification.

### 2.4. PsGrx Interacts with GR and AhpC

The results of BiFC are illustrated in Figure 6, the fluorescence signal was observed when GR and AhpC fused with the C-terminal of YFP and PsGrx fused with the N-terminal of YFP (PsGrx-YN) were transiently co-expressed. In controls, no fluorescence signal was observed when PsGrx-YN and target protein-YC were co-expressed with the YN and YC empty vector, respectively. These results demonstrated that PsGrx physically interacted with GR and AhpC. However, there is no interaction between PsGrx and four other targets. In parallel, we conducted a Co-IP assay using co-expressed PsGrx and target proteins in *Nicotiana benthamiana*. As expected, the results verified the interaction between PsGrx and GR, PsGrx, and AhpC (Figure 7). Together, these results clearly demonstrated that PsGrx directly interacted with GR and AhpC. Therefore, these two target proteins were selected for further analyses.

To assess whether GR and AhpC were regulated by PsGrx in response to low-temperature stress, the level of *gr* and *ahpc* expression was quantified in wild-type ANT206 and mutant Δ*psgrx*. qRT-PCR analysis was performed to illustrate the expression patterns of target proteins under low temperature. As shown in Figure 8, the enhanced expression of *psgrx* in the 5 °C treatment demonstrated that the expression of *psgrx* was significantly induced by low temperature. The expression of *gr* and *ahpc* was significant enhanced dependent of temperature, indicating that *gr* and *ahpc* were also sensitive to temperature. In addition, *gr* and *ahpc* expression were lower in the mutant Δ*psgrx* than in the wild type, which indicated that *gr* and *ahpc* was regulated by *psgrx*. These results demonstrated that the expressions of *psgrx*, *gr*, and *ahpc* were all induced by temperature, and *psgrx* enhanced the expression of *gr* and *ahpc*.

### 2.5. PsGrx Is Participated in Glutathione Metabolism by Enhancing GR Activity

GR participates in the ascorbate-glutathione cycle which involves the antioxidant metabolites, such as ascorbate, glutathione, NADPH, and the enzymes linking these metabolites. It is worth noting that GR is one of glutathione metabolism parameters [39]. To analyze whether PsGrx is participated in glutathione metabolism, the activity of GR was measured in wild-type and mutant Δ*psgrx*. Under low-temperature suffering, the activity of GR was decreased (Figure 9), while the GR activity in *Phalaenopsis* seedlings was induced by low temperature [40]. GR activity was higher in WT than mutant Δ*psgrx*, demonstrating that PsGrx could enhance GR activity. Furthermore, *psgrx* enhanced the expression of *gr* (Figure 8). Therefore, PsGrx might participate in glutathione metabolism by enhancing the activity and expression of GR at low temperature.

### 2.6. PsGrx Is Involved in ROS-Elimination Pathway by Regulating AhpC

Analysis of genes from the oxidative stress-defense pathway encoding thioredoxin reductase (TrxR) is able to indicate whether they are participating in ROS elimination [41]. Since AhpC is involved in the ROS-elimination pathway [41] and interacts with PsGrx during the response of ANT206 to low temperature, we hypothesized that PsGrx might affect the function of AhpC in this process. To test this hypothesis, in-frame deletion mutant Δ*ahpc* (450 bp) and double deletion mutant Δ*psgrx* Δ*ahpc*, which demonstrated good hereditary stability, were constructed (Figure 10). Subsequently, we analyzed the expression of *trxR* in wild-type ANT206, mutant Δ*psgrx*, Δ*ahpc*, and Δ*psgrx* Δ*ahpc* (Figure 11). The expression levels of *trxR* in wild-type and mutant were increased at 5 and 10 °C, indicating that the expression of *trxR* was induced by low temperature. The expression of both *trxR* in *Apis mellifera* L. and *Apis cerana* F. rapidly also increased after exposure to 4°C, with a stronger effect induced by cold stress [42]. At 5 °C, the expression of *trxR* in mutant Δ*psgrx* and Δ*ahpc* was higher than Δ*psgrx* Δ*ahpc*, which demonstrated that *ahpc* and *psgrx* both possessed the ability to enhance *trxR* expression. Similarly, *ahpc* in *Bifidobacterium longum* strain NCC2705 also increased the expression of *trxR* [41]. Importantly, the *trxR* expression in Δ*psgrx*, Δ*ahpc*, and wild-type ANT206 showed 3.7, 2.4, and 10-fold more than mutant Δ*psgrx* Δ*ahpc* at 5 °C, respectively, the *trxR* expression fold in wild-type ANT206 was higher than the sum of the other two deletion mutants. The possible reason for this phenomenon was that PsGrx interacted with AhpC, which increased the expression of *trxR*. Therefore, PsGrx could increase the expression of *trxR* by regulating AhpC.

The hypothetical regulation pattern of PsGrx in cells is illustrated in Figure 12. Oxidative stress, arising from excessive accumulation of ROS, can be induced by low temperature. Deletion *psgrx* impaired low-temperature growth of mutant Δ*psgrx* and *psgrx* was considered to be involved in low-temperature regulation and positively regulated the response to low temperature to ANT206 (Figure 3). Similar mechanisms have been described in *Saccharomyces cerevisiae*; mutants of the genes involved on the main antioxidant response pathways were constructed, and a comparison of the µ_max_ of the mutants with each parental strain under low temperature was also illustrated [24]. PsGrx interacted with GR and AhpC, *psgrx* was capable of enhancing the expression of *gr* and *ahpc* (Figure 8). Furthermore, PsGrx could be involved in glutathione metabolism by enhancing the expression and activity of GR (Figure 9); the interaction between PsGrx and AhpC enhanced the expression of *trxR*, indicating that PsGrx might participate in the ROS-elimination pathway by regulating AhpC (Figure 11) to protect ANT206 from low-temperature stress. 

## 3. Conclusions

Low temperature is a major stress that adversely affects microbial growth in the Antarctic, and unraveling the adaptation mechanisms of Antarctic microorganisms has always been a matter of interest. In this study, Antarctic psychrophile *Psychrobacter* sp. ANT206 was used as materials; deletion mutation, yeast two-hybrid, and qRT-PCR were used to study the function of PsGrx. The results showed that *psgrx* improved the tolerance of ANT206 to low temperature. In addition, several target proteins that interacted with PsGrx were screened and identified. Among the target proteins, GR and AhpC were regulated by PsGrx under low temperature. Taken together, the data PsGrx may participate in glutathione metabolism and the ROS-elimination pathway by regulating GR and AhpC under low temperature to improve the growth of ANT206. Studying the regulatory function of PsGrx would provide valuable insights into understanding complex cellular physiologies such as stress responses. The findings also provide a novel understanding of low-temperature adaptation in microorganisms.

## 4. Materials and Methods

### 4.1. Strains and Material

The wild-type strain *Psychrobacter* sp. ANT206 was isolated from the Antarctic sea-ice. ANT206, the deletion mutant Δ*psgrx*, Δ*ahpc* and double deletion mutant Δ*psgrx* Δ*ahpc* were all cultured in 2216E medium under environmental conditions of 15 °C and pH 7.5. At 37 °C, auxotrophic *Escherichia coli* (*E. coli*) WM3064 and *E. coli* WM3064 that contained the suicide plasmid pRE112 were cultured in the LB medium that had a pH value of 7.0 and contained meso-2,6-diaminopimelic acid (DAP) of 50 μg/mL. *E. coli* S17-1, and *E. coli* S17-1 that contained the suicide plasmid pDS132 were cultured in the LB medium. The above strains were maintained in our laboratory. Strain AH109, Y2H, and related plasmids used in a Y2H assay were purchased from the Beijing Genomics Institute (BGI, Beijing, China). All other reagents were acquired from Sinopharm (Beijing, China), and were of analytical grade or higher.

### 4.2. Construction of Mutant Strain Δpsgrx, Δahpc and Δpsgrx Δahpc

To study the function of PsGrx, a deletion mutant of *psgrx* was constructed using the allele replacement method [43]. Taking into account the expression of genes, we deleted a DNA fragment (90 bp) in the *psgrx*. Using the whole genome as a template, the primers Δ*psgrx*-P1, which has a *X*baI site, and Δ*psgrx*-P2 were used to amplify the 1–87 bp of the deletion mutant. The amplification products were labeled *grx*1. Using Δ*psgrx* -P3 and P4 as primers, the method above was repeated to amplify 178–264 bp fragment, and the amplification products were labeled *grx*2. Fragments *grx*1 and *grx*2 were fused via PCR and named Δ*psgrx*. Plasmid pRE112 and Δ*psgrx* were ligated together after being digested separately. This experiment chose the suicide plasmid pRE112 as the carrier and transferred the ligation products into the plasmid. After that, the ligation products were mated with auxotrophic *E. coli* WM3064 and entered the strain ANT206. By homologous recombination, this experiment integrated the transconjugants into the genome of ANT206. At 15 °C, the target transconjugants were then selected by the use of the 2216E solid medium that had kanamycin and DAP of 50 μg/mL. Next, a double-crossover recombination fragment was incubated in 2216E solid medium with 10% sucrose at 15 °C to culture the mutant Δ*psgrx*. Mutant Δ*ahpc* was constructed via the same method, a DNA fragment (102 bp) in the *ahpc* was deleted, and Δ*ahpc*-P1 P2 and Δ*ahpc*-P3 P4 were used to amplify the 1–246 and 349–552 bp of the deletion mutant, respectively. Then, fragments were fused via PCR and named Δ*ahpc*. The ligation suicide plasmid pDS132 and Δ*ahpc* were mated with *E. coli* S17-1 (named *E. coli* S17-1/pDS132-Δ*ahpc*) and entered the strain ANT206 by homologous recombination. Then, target transconjugants were selected by 2216E solid medium that had chloramphenicol of 50 μg/mL. Furthermore, based on mutant strain Δ*psgrx*, double deletion mutant Δ*psgrx* Δ*ahpc* was obtained by mating *E. coli* S17-1/pDS132-Δ*ahpc* and mutant strain Δ*psgrx*. Finally, mutant strain Δ*psgrx*, Δ*ahpc,* and Δ*psgrx* Δ*ahpc* were sequenced by The Beijing Genomics Institute (BGI, Beijing, China). Kanamycin or chloramphenicol resistance stability and the genetic stability of mutants were determined as previously described [44]. The primers used in allele replacement are listed in Table 2, and the cultured mutant strains were analyzed and identified via PCR reaction.

### 4.3. Low-Temperature Treatment, MDA Activity and Protein Carbonylation Assay

To investigate the effects of low temperature on wild-type ANT206 and mutant strains, ANT206, mutant Δ*psgrx*, Δ*ahpc* and Δ*psgrx.* Δ*ahpc* were added to the fresh 2216E media, until the density of the cells at 600 nm (OD_600_) achieved 0.05. ANT206 and mutant strains were incubated at 5 °C, 10 °C and 15 °C for 72 h. The OD_600_ of ANT206 and mutant Δ*psgrx* was determined via spectrophotometry (UV2000, Shimazu, Japan). The growth parameter maximum specific growth rate (μ_max_) of ANT206 and mutant Δ*psgrx* was calculated from each treatment by directly fitting OD measurements versus time to the reparametrized Gompertz equation proposed by Zwietering et al. [45]. Another growth parameter generation time (GT) was calculated based on the previous method [46]. The MDA levels, protein carbonylation content, and GR activity were determined using commercial kits (Nanjing Jiancheng Bioengineering Institute, Nanjing, China) and detected via spectrophotometry. Moreover, through the Bradford Protein Assay Kit (Biyuntian, Haimen, China), this study determined protein concentrations in the cell extract using bovine serum albumin as standard.

### 4.4. cDNA Library of Strain ANT206

To test the interaction between PsGrx and proteins from strain ANT206, this experiment used *psgrx* as a bait and the cDNA library of the ANT206 strain as prey to carry out a genome-wide Y2H screening. With the intention of preparing the cDNA library, TRIzol^®^ Reagent (Ambion, Carlsbad, CA, USA) was used to extract the total RNA was extracted from strain ANT206, and the Oligotex^®^ mRNA Mini Kit (Qiagen, Hilden, Germany) was employed to isolate the mRNA. After that, Make Your Own Mate & Plate^®^ Library System (Clontech Laboratories Inc., Palo Alto, CA, USA) was used to reverse-transcribe the first-strand cDNA from the mRNA according to the guidelines of the manufacturer and employed the amplified double-stranded cDNA to conduct the assessment [47].

### 4.5. Yeast Two-Hybrid Analysis

To identify the interacting partners of PsGrx, the Y2H screen was performed with the Matchmaker GAL4 Two-hybrid System 3 (Clontech Laboratories Inc., Palo Alto, CA, USA). The schematic figure of the Y2H process was illustrated in Figure 13. To test bait auto-activation, we marked the pGADT7-largeT and pGBKT7-p53 as positive controls, pGADT7-largeT and pGBKT7-laminC as negative controls, and pGADT7 and pGBKT7-PsGrx as the experimental group. Plasmids were transformed into the AH109 strain and cultured on selective medium (SD-TL) at 30 °C for 4 days. Six transformants were randomly selected and transferred to two separate selective media plates at 30 °C for culturing for 4 days. The bait yeast culture and prey yeast cDNA library were gently mixed. A total of 30 transformant colonies that grew again were selected and transferred to the SD-TL and SD-TLHA+X-α-Gal plates at 30 °C for culturing for 3 days. Finally, we sequenced the colonies that grew on both plates and determined the gene names using BLAST (https://blast.ncbi.nlm.nih.gov/Blast.cgi, last accessed on: 21 January 2022).

### 4.6. Bimolecular Fluorescence Complementation (BiFC) and Co-Immunoprecipitation (Co-IP) Assay

Since this study’s emphasis is stress-related reaction targets and new targets, the interactions between PsGrx and these six proteins (GR, glutathione peroxidase, AhpC, GspI, DNA translocase FtsK, and RNase E) were further analyzed. With the intention of generating constructs for BiFC assays, this experiment amplified full-length cDNA fragments of these six proteins with PCR methods (the primers are listed in Table 3) and subcloned them into the pDONR221 vector and then recombined them into the YN (pEarleyGate201-YN) and YC (pEarleyGate202-YC) vectors [48]. After that, this experiment introduced the constructs to the *Agrobacteriumtumefaciens* strain GV3101. Then, a 1 mL needleless syringe was used to co-infiltrate them into *Nicotiana benthamiana* leaves’ abaxial side. The *Nicotiana benthamiana* leaves used in this experiment should be 4 to 6 weeks old. Infected tissues were examined after they were infiltrated for 48 h. This experiment adopted the Confocal Spectral Microscope Imaging System (Leica TCS SP5, Wetzlar, Germany), to capture the YFP fluorescence, setting an argon blue laser at 488 nm, a beam splitter at 500 nm for excitation, as well as a spectral detector between 515 and 540 nm. Based on the BiFC experiment, a Co-Ip assay was used for analysis, and PsGrx with a hemagglutinin (HA) tag, GR, and AhpC with a Flag tag were co-expressed in *Nicotiana benthamiana*. Proteins were extracted using lysis buffer and DTT after 48 h incubation. Then, 10 μL of anti-HA-tag magnetic bead buffer was added and the samples were incubated for 3 h at 4 °C to immunoprecipitate the proteins. Next, Western blotting was performed to transfer the proteins to the PVDF membrane. Anti-HA and anti-Flag with sodium azide were dissolved and added to the PVDF membrane, incubating for 3 h.

### 4.7. RNA Extraction and Quantitative Real-Time PCR

The expressions of *psgrx* and target proteins were determined by performing qRT-PCR at 5 °C, 10 °C, and 15 °C. Total RNA was extracted from ANT206 and the mutant Δ*psgrx*, Δ*ahpc* and Δ*psgrx* Δ*ahpc* cultured in each different temperature environment using TRIzol^®^ Reagent (Ambion, Carlsbad, CA, USA), which was followed by centrifuging at 12,000× *g* for 5 min at 4 °C. The chloroform and isopropanol were added, which was followed by centrifuging at 12,000× *g* for 5 min at 4 °C. RNA precipitate was obtained from the bottom of the tube. The extracted RNA was exposed to RNase-free DNase to remove any residual genomic DNA that may present in the RNA. The qRT-PCR was performed by a PCR instrument (Applied Biosystems 7500, Carlsbad, America). A 16S rRNA sequence of strain ANT206 was used as the internal reference for normalizing gene expression. The comparative C_t_ (2^−ΔΔCt^) method was used to calculate relative gene expression [49]. The primers used for qRT-PCR are listed in Table 4. 

### 4.8. Statistical Analysis

Statistical significance of the results was analyzed by Statistical Product and Service Solutions (SPSS) 22.0 software. Data are presented are means ± SD from three independent experiments, asterisks indicate significant differences. The differences were considered to be significant if *p* < 0.05 and were indicated by one asterisk, those at *p* < 0.01 were indicated by double asterisks.

## Figures and Tables

**Figure 1 ijms-23-01313-f001:**
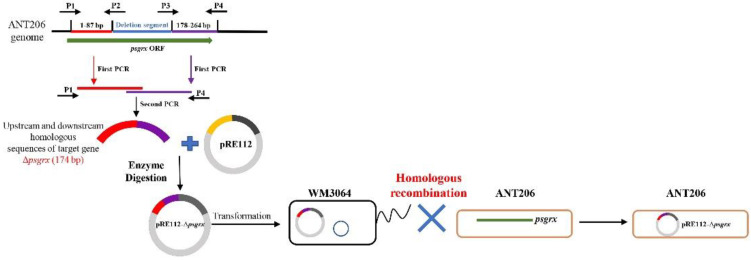
Schematic diagram of the in-frame gene deletion mutant strain construction process of *psgrx* of ANT206.

**Figure 2 ijms-23-01313-f002:**
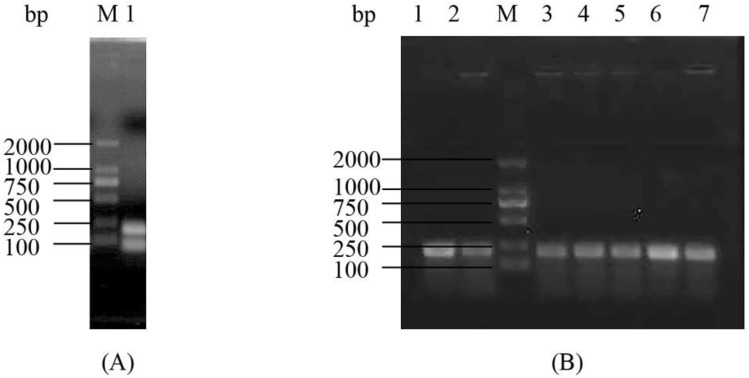
(**A**) Electrophoresis detection of the mutant Δ*psgrx* fragment. 1, PCR product of mutant Δ*psgrx* fragment; M, DL2000 DNA marker; (**B**) The colony electrophoresis detection of the subcultured mutant Δ*psgrx*. M, DL2000 DNA marker; 1, Δ*psgrx* fragment; 2–7, PCR products of 5, 10, 15, 20, 25, and 30 generations colonies, respectively.

**Figure 3 ijms-23-01313-f003:**
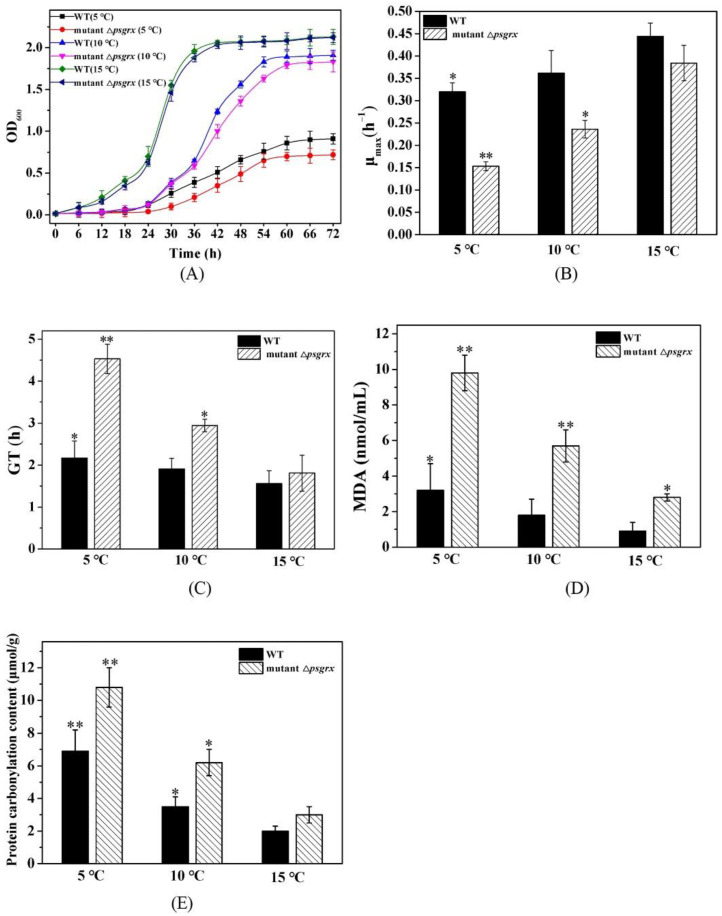
(**A**) The growth curves of wild-type ANT206 (WT) and mutant Δ*psgrx* under 5, 10, and 15 °C. (**B**) The maximum specific growth rates (μmax) of WT and mutant Δ*psgrx* under 5, 10, and 15 °C. Asterisks indicate significant differences compared with WT at 15 °C (* *p* < 0.05; ** *p* < 0.01). (**C**) The generation time (GT) of WT and mutant Δ*psgrx* under 5, 10, and 15 °C. Asterisks indicate significant differences compared with WT at 15 °C (* *p* < 0.05; ** *p* < 0.01). (**D**) Effect of MDA level of WT and mutant Δ*psgrx* under 5, 10, and 15 °C. Asterisks indicate significant differences compared with WT strains at 15 °C (* *p* < 0.05; ** *p* < 0.01). (**E**) Effect of protein carbonylation content of WT and mutant Δ*psgrx* under 5, 10, and 15 °C. Asterisks indicate significant differences compared with WT at 15 °C (* *p* < 0.05; ** *p* < 0.01).

**Figure 4 ijms-23-01313-f004:**
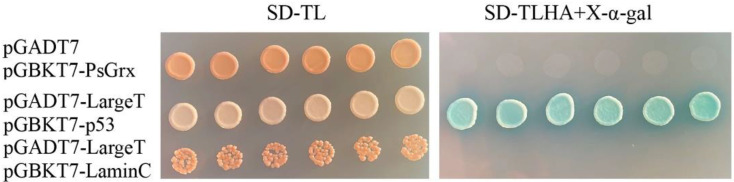
Determination of the auto-activation activity of PsGrx baits in yeast cells. The colonies growing in the SD-TL (SD/-Leu/-Trp) agar plate indicate that the plasmid is successfully introduced into yeast cells, and the blue colonies that can grow in the SD-THLA (SD/-Ade/-His/-Leu/-Trp)+X-α-Xal agar plate can be auto-activated. pGADT7-LargeT + pGBKT7-p53 was negative control, and pGADT7-LargeT and pGBKT7-LaminC was positive control.

**Figure 5 ijms-23-01313-f005:**
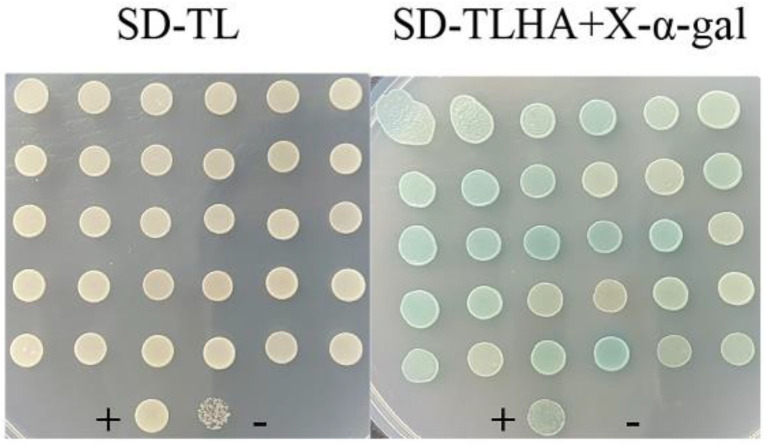
Positive Y2H interaction between PsGrx, as bait, and cDNA library of ANT206 as prey. In both Y2H assays, positive interactions are indicated by cell color on TL (SD/-Leu/-Trp) and TLHA (SD/-Ade/-His/-Leu/-Trp) + X-α-Xal agar plates. “+”: positive; “-”: negative.

**Figure 6 ijms-23-01313-f006:**
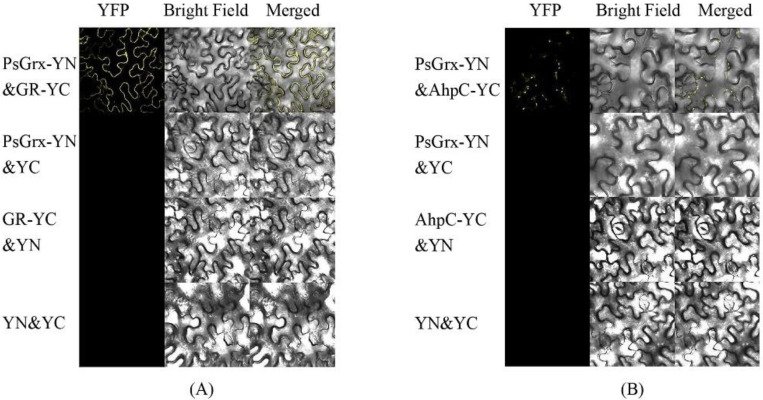
Bimolecular fluorescence complementation assay of the interaction between PsGrx and GR (**A**), PsGrx and AhpC (**B**) in *Nicotiana benthamiana* leaves. PsGrx, GR and AhpC were fused with N- or C-terminal of YFP in vector, respectively. PsGrx-YN&YC, GR-YC&YN, AhpC-YC&YN and YN&YC were used as the negative controls.

**Figure 7 ijms-23-01313-f007:**
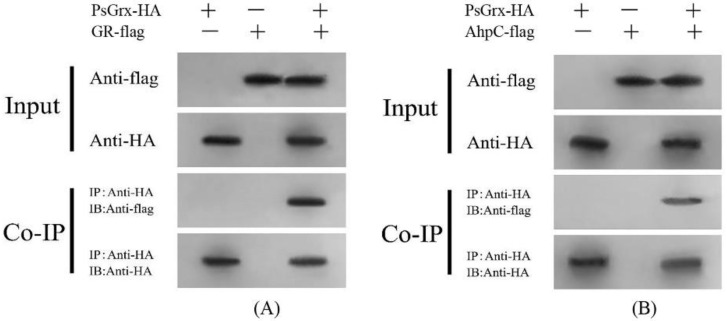
Interaction of PsGrx and GR or AhpC in co-immunoprecipitation assays. PsGrx with a hemagglutinin (HA) tag, and (**A**) GR or (**B**) AhpC with a Flag tag co-expressed in *Nicotiana benthamiana*, which were immunoprecipitated using anti-HA and analyzed by protein gel blot analysis.

**Figure 8 ijms-23-01313-f008:**
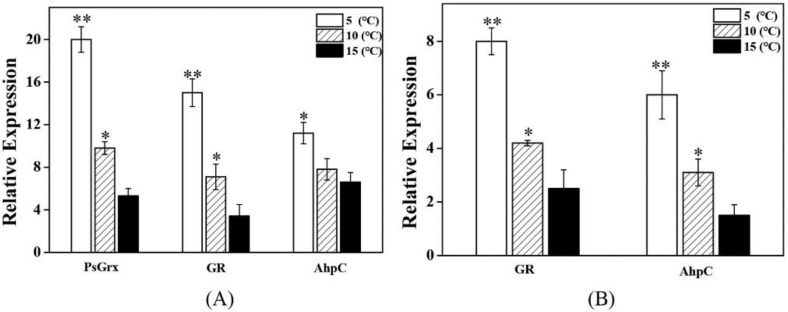
qRT-PCR analysis of *psgrx* and target proteins expression under low temperature in (**A**) wild-type ANT206 (WT) and (**B**) mutant Δ*psgrx*. The expression of *psgrx*, *gr* and *ahpc* is a multiple of the 16S rRNA expression of the internal reference gene. Data presented are means ± SD from three independent experiments; asterisks indicate significant differences compared with strains at 15 °C (* *p* < 0.05; ** *p* < 0.01).

**Figure 9 ijms-23-01313-f009:**
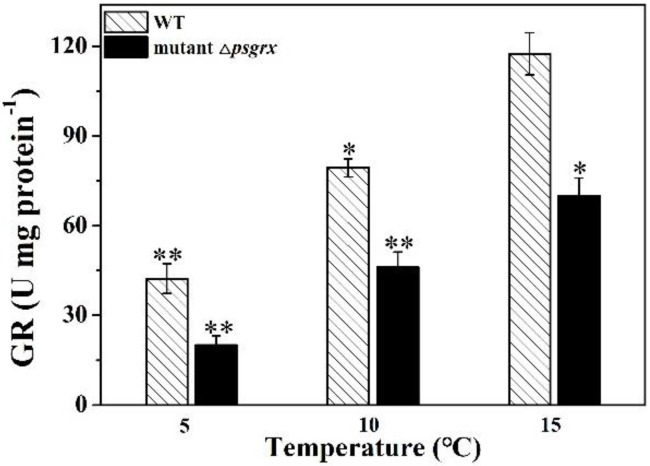
The activity of GR under low temperature in wild-type ANT 206 (WT) and mutant Δ*psgrx*. Data presented are means ± SD from three independent experiments, asterisks indicate significant differences compared with WT strains at 15 °C (* *p* < 0.05; ** *p* < 0.01).

**Figure 10 ijms-23-01313-f010:**
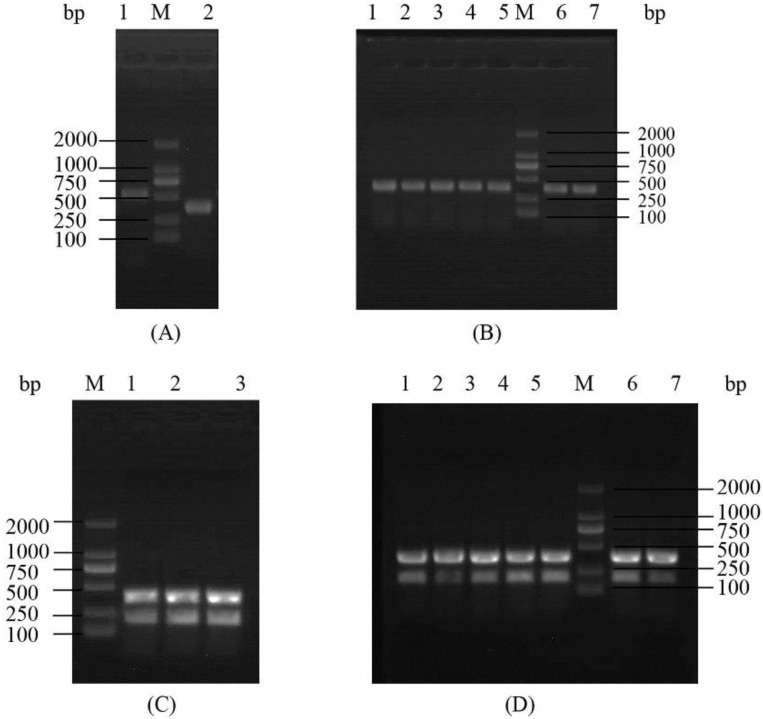
(**A**) The electrophoresis detection of the mutant Δ*ahpc* fragment. 1, PCR product of *ahpc* fragment; 2, PCR product of mutant Δ*ahpc* fragment; M, DL2000 DNA marker. (**B**) The colony electrophoresis detection of the subcultured mutant Δ*ahpc*. M, DL2000 DNA marker; 1, Δ*ahpc*; 2–7, PCR products of 5, 10, 15, 20, 25 and 30 generations colonies, respectively. (**C**) The electrophoresis detection of the double deletion mutant Δ*psgrx* Δ*ahpc* fragment. 1–3, PCR product of mutant Δ*psgrx* Δ*ahpc* fragment; M, DL2000 DNA marker. (**D**) The colony electrophoresis detection of the subcultured mutant Δ*psgrx* Δ*ahpc*. M, DL2000 DNA marker; 1, Δ*psgrx* Δ*ahpc*; 2–7, PCR products of 5, 10, 15, 20, 25 and 30 generations colonies, respectively.

**Figure 11 ijms-23-01313-f011:**
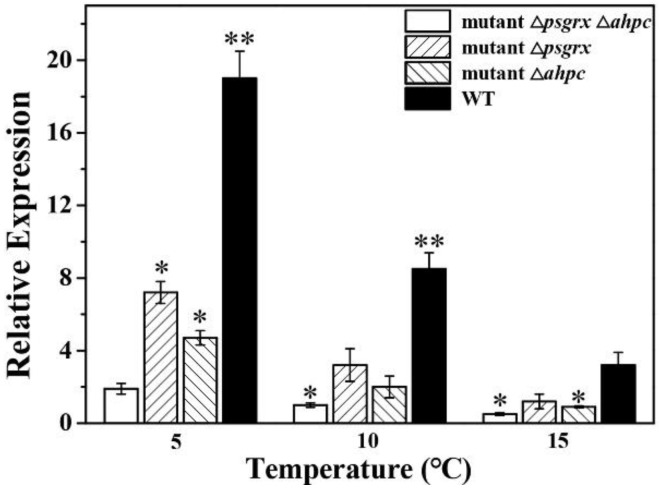
qRT-PCR analysis of *trxR* expression under low temperature in wild-type ANT206 (WT), mutant Δ*psgrx*, Δ*ahpc* and Δ*psgrx* Δ*ahpc*. The expression of *trxR* is a multiple of the 16S rRNA expression of the internal reference gene. Data presented are means ± SD from three independent experiments; asterisks indicate significant differences compared with WT at 15 °C (* *p* < 0.05; ** *p* < 0.01).

**Figure 12 ijms-23-01313-f012:**
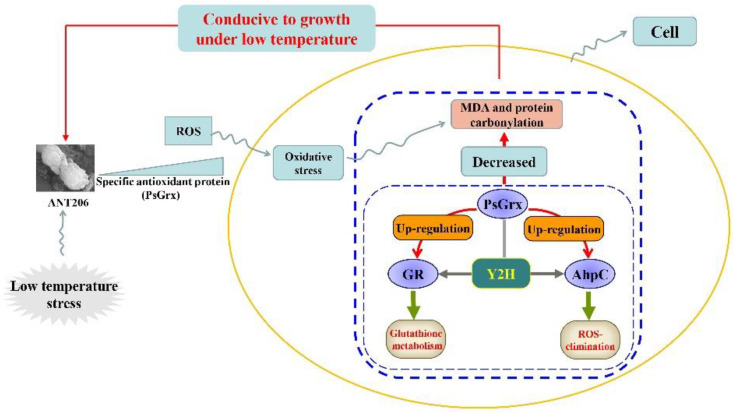
Hypothetical regulation pattern of PsGrx in cells under low temperature.

**Figure 13 ijms-23-01313-f013:**
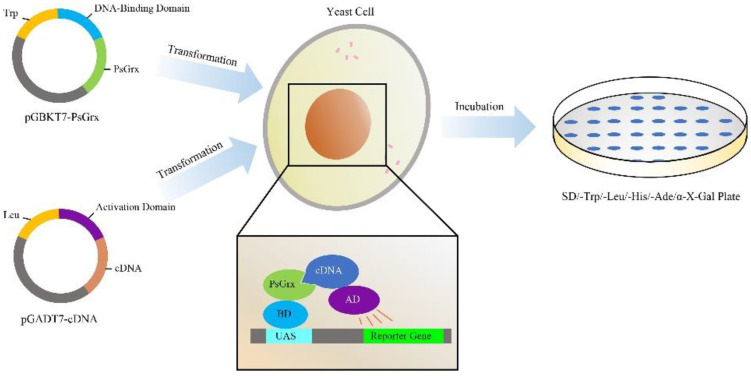
Schematic figure of the Y2H process.

**Table 1 ijms-23-01313-t001:** Putative interaction proteins identified by Y2H assay.

Gene Number	Gene Name	Functional Class
1	Glutathione reductase	Stress-related reactions
2	Glutathione peroxidase	Stress-related reactions
3	Alkyl hydroperoxide reductase	Stress-related reactions
4	DNA photolyase	DNA modification
5	Aldehyde dehydrogenase	Biosynthesis
6	Translation elongation factor	Translation
7	Nucleoside diphosphate kinase	Translation
8	Glyceraldehyde 3-phosphate dehydrogenase	Glycolysis
9	Transketolase	Calvin cycle; associated reactions
10	Fructose-1,6-bisphosphatase	Calvin cycle; associated reactions
11	Triosephosphate isomerase	Calvin cycle; associated reactions
12	Fructose-bisphosphate aldolase	Calvin cycle; associated reactions
13	Phosphoglycerate kinase	Calvin cycle; associated reactions
14	Carbonic anhydrase	Calvin cycle; associated reactions
15	Methionine synthase	Sulfur metabolism
16	Cysteine synthase	Sulfur metabolism
17	Aminotransferase	Nitrogen metabolism
18	GspI	Protein secretion
19	DNA translocase FtsK	DNA transportation; cell division
20	RNase E	RNA metabolism; transcription
21	AarF/Abc1/UbiB kinase family protein	Fatty acid metabolism; protein synthesis
22	peptidoglycan-binding protein LysM	Cell separation
23	ATP synthase α chain	ATP metabolism
24	Methyltransferase small domain	Hypothetical protein
25	Thioesterase	Hypothetical protein
26	Transglutaminase-like domain	Hypothetical protein

**Table 2 ijms-23-01313-t002:** Primers used for the construction of the mutant Δ*psgrx* and Δ*ahpc* (underlines represented cleavage sites of *X*baI and *S*acI, respectively).

Name	Primer Sequences	Restriction Enzyme Cutting Sites
Δ*psgrx*-P1	5′-GCTCTAGACGATGACTGTATCTGTTAAAG-3′	*X*baI
Δ*psgrx*-P2	5′-CGGTGCGATAGTTATTCTCTTCATAATC-3′	
Δ*psgrx*-P3	5′-GAAGAGAATAACTATCGCACCGTGC-3′	
Δ*psgrx*-P4	5′-TAGAGCTCTTAACCCGCTAATAGCTC-3′	*S*acI
Δ*ahpc*-P1	5′-ATGTCTAGAATGACGACTGATAGCG-3′	*X*baI
Δ*ahpc*-P2	5′-GTTCTCAATGTGACCCCAAAAATAG-3′	
Δ*ahpc*-P3	5′-CAAGAGTTACACAGATAAAAACCCC-3′	
Δ*ahpc*-P4	5′-TCAGAGCTCTAAAAACTGACGACAG-3′	*S*acI

**Table 3 ijms-23-01313-t003:** Primers used for BiFC.

Name	Primer Sequences
*psgrx*-F	5′-ACAAGTTTGTACAAAAAAATGACTGTATCTGTTAAAGTTTATAC-3′
*psgrx*-R	5′-CACCACTTTGTACAAGAAACCCGCTAATAGCTCGTCAAG-3′
*gr*-F	5′-ATGACAAAACATTATGATTATATTT TCCATTGGCGGC-3′
*gr*-R	5′-CTAACGCATCGTCACAAACTCTTCTGAGCCAGTTGGATGAAT-3′
*ahpc*-F	5′-ATGACGACTGATAGCGACAAGACGACTGAGAGATCTAAAAAG-3′
*ahpc*-R	5′-AAAAACTGACGACAGCCACAATCTTAATTTCAATGACCATAAC-3′
*glutathione peroxidase*-F	5′-ATGACTACTATTTATGATTTTAGTGCTGAGCGTATGGCAT-3′
*glutathione peroxidase*-R	5′-TTTGCACGCCTCCTTAACTTGGTCAAGATCAGGGCTGAAC-3′
*gspI*-F	5′-ATGATAAATAATGACAGAGCCAAACCTAACCATGTAAACCGA-3′
*gspI*-R	5′-GTTTGGCTCTGTCATTATTTATCATTTCGGTTTACATGGTTAG-3′
*ftsK*-F	5′-GTGATATCAGCACCAATTATTGATTACTTAAAAAAGGGCATA-3′
*ftsK*-R	5′-AATCAATAATTGGTGCTGATATCACATATGCCCTTT-3′
*Rnase E*-F	5′-ATGAAACGCATTTTAATCAACGCCACCCAAAACGAAGAAATTC-3′
*Rnase E*-R	5′-GCTCTCTCTATCTGAGTTATCTGAATCATCTGACTCTAGTT-3′

**Table 4 ijms-23-01313-t004:** Primers used for qRT-PCR.

Name	Primer Sequences
*psgrx*-F	5′-GGCGTTGATTATGAAGAGATTGGCATG-3′
*psgrx*-R	5′-TGTGGCACGGTACGATAGTTATTAGTC-3′
*gr*-F	5′-TGTATGTCCGTCAGCACTCG-3′
*gr*-R	5′-TCGCCCAAATCAAGCAGTCT-3′
*ahpc*-F	5′-CAAGTCCGGCTCTGACCAAG-3′
*ahpc*-R	5′-CTTGGCTCATCTCGCCATCT-3′
*trxR*-F	5′-CTGATCGTCAACAGCGGTCT-3′
*trxR*-R	5′-CAGCAGAGGTGATCGCTTGA-3′
16S-F	5′-CCTTCGCCATCGGTATTCCTCCAG-3′
16S-R	5′-GAGCTAGAGTATGTGAGAGG-3′

## Data Availability

The datasets used and/or analyzed during the current study are available from the corresponding authors on reasonable request (wangquanfuhit@hit.edu.cn).

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
