# Peer review of "Glutaredoxin Interacts with GR and AhpC to Enhance Low-Temperature Tolerance of Antarctic Psychrophile Psychrobacter sp. ANT206"

_ijms, 2022, doi:10.3390/ijms23031313_

Round 1
Reviewer 1 Report
The introduction is not enough to have a good picture of the topic and some references in other organims are missing such as yeast while huge emphasis is given to plants.
Figure 2a: the growth curves should be analyzed in order to have the growth parameters such as maximum specific growth rate and perform statistics.
Figure 3b and c: I think that something is wrong with the statistics because at a glance it is obvious that the differences at 15C in panel B and probably at 10 in panel C are not indicated. The same is visible in figure 7 and along the manuscript.
Figure 6: the legend should be more complete
Figure 7 and 11: relative expression regarding what constitutive gene?
Similar mechanisms have been described in Saccharomyces cerevisiae, authors should include and discuss also the results (Garcia-Rios et al.,2016).
Conclusions section should be placed after results and discussion section.
A statistical section should be included in the mat&met section describing the applied test.
It would be usefull to have a schematic figure of the Y2H process used by the authors
Author Response
Reviewer 1
The introduction is not enough to have a good picture of the topic and some references in other organims are missing such as yeast while huge emphasis is given to plants.
Response: We appreciate very much for the Reviewer’s good comments. We have added 4 additional other organisms Grx references and deleted 2 plant Grx references to the Introduction. This part has been revised, please see line 42-51.
Figure 2a: the growth curves should be analyzed in order to have the growth parameters such as maximum specific growth rate and perform statistics.
Response: We highly agree for the Reviewer’s useful comments. We have added two growth parameters (maximum specific growth rate and generation time) in the revised version. This part has been added, please see line 100-108.
Figure 3b and c: I think that something is wrong with the statistics because at a glance it is obvious that the differences at 15C in panel B and probably at 10 in panel C are not indicated. The same is visible in figure 7 and along the manuscript.
Response: This part has been revised, please see Figure 3, Figure 8 and Figure 11.
Figure 6: the legend should be more complete.
Response: This part has been added, please see Figure 6 legend.
Figure 7 and 11: relative expression regarding what constitutive gene?
Response: Expression of relative internal reference gene 16S rRNA of ANT206. This part has been added, please see of Figure 8 and Figure 11 legends.
Similar mechanisms have been described in Saccharomyces cerevisiae, authors should include and discuss also the results (Garcia-Rios et al.,2016).
Response: We highly agree and appreciate very much for the Reviewer’s nice comments. This part has been added, please see line 105-107 and line 277-280.
Conclusions section should be placed after results and discussion section.
Response: This part has been modified.
A statistical section should be included in the mat&met section describing the applied test.
Response: This section has been added in this version, please see section 4.8.
It would be usefull to have a schematic figure of the Y2H process used by the authors
Response: We highly agree and appreciate very much for the Reviewer’s nice comments. This part has been added, please see Figure 13.
Reviewer 2 Report
The Quanfu Wang and coworkers performed a construction of an in-frame deletion mutant of psgrx (Δpsgrx). Further, two target proteins, glutathione reductase (GR) and alkyl hydroperoxide reductase subunit C (AhpC), were
regulated by PsGrx under low temperature, and the interactions were confirmed via bimolecular fluorescence complementation and co-immunoprecipitation. The initial results look promising. The authors have also competently perfomed the studies. I do recommend an acceptance for this manuscript.
Author Response
Reviewer 2
The Quanfu Wang and coworkers performed a construction of an in-frame deletion mutant of psgrx (Δpsgrx). Further, two target proteins, glutathione reductase (GR) and alkyl hydroperoxide reductase subunit C (AhpC), were regulated by PsGrx under low temperature, and the interactions were confirmed via bimolecular fluorescence complementation and co-immunoprecipitation. The initial results look promising. The authors have also competently perfomed the studies. I do recommend an acceptance for this manuscript.
Response: We highly appreciate very much for the Reviewer’s comments.